# Geographical Authentication of *Macrohyporia cocos* by a Data Fusion Method Combining Ultra-Fast Liquid Chromatography and Fourier Transform Infrared Spectroscopy

**DOI:** 10.3390/molecules24071320

**Published:** 2019-04-03

**Authors:** Qin-Qin Wang, Heng-Yu Huang, Yuan-Zhong Wang

**Affiliations:** 1Institute of Medicinal Plants, Yunnan Academy of Agricultural Sciences, Kunming 650200, China; wqq6501@163.com; 2College of Traditional Chinese Medicine, Yunnan University of Traditional Chinese Medicine, Kunming 650500, China

**Keywords:** *Macrohyporia cocos*, data fusion, liquid chromatography, fourier transform infrared spectroscopy, partial least squares discriminant analysis, authentication

## Abstract

*Macrohyporia cocos* is a medicinal and edible fungi, which is consumed widely. The epidermis and inner part of its sclerotium are used separately. *M. cocos* quality is influenced by geographical origins, so an effective and accurate geographical authentication method is required. Liquid chromatograms at 242 nm and 210 nm (LC_242_ and LC_210_) and Fourier transform infrared (FTIR) spectra of two parts were applied to authenticate the geographical origin of cultivated *M. cocos* combined with low and mid-level data fusion strategies, and partial least squares discriminant analysis. Data pretreatment involved correlation optimized warping and second derivative. The results showed that the potential of the chromatographic fingerprint was greater than that of five triterpene acids contents. LC_242_-FTIR low-level fusion took full advantage of information synergy and showed good performance. Further, the predictive ability of the FTIR low-level fusion model of two parts was satisfactory. The performance of the low-level fusion strategy preceded those of the single technique and mid-level fusion strategy. The inner parts were more suitable for origin identification than the epidermis. This study proved the feasibility of the data fusion of chromatograms and spectra, and the data fusion of different parts for the accurate authentication of geographical origin. This method is meaningful for the quality control of food and the protection of geographical indication products.

## 1. Introduction

The dried sclerotium of *Macrohyporia cocos*, belonging to Polyporaceae, is an herbal medicine (called Poria) that can be used as food, and has been approved by the National Health Commission of the People’s Republic of China. It plays an indispensable role in numerous drugs, such as the liquid oral formulation of *Poriacocos* polysaccharides, Sijunzi Tang, Liuwei Dihuang Wan and Chuanbei Pipa Gao. Various kinds of Poria-based foods and skin cosmetics such as sleep-friendly tea, Tuckahoe pie, Guiling jelly (drinks made from turtle shell and medicinal herbs), Guiling jelly soft candy and the Poria facial mask, are pretty popular. Present phytochemical investigation suggests that this fungus contains terpenes and polysaccharides, which present beneficial biological properties, such as a prebiotic effect, through the modulation of gut microbiota composition [1], anti-hyperlipidemic [2], anti-cancer [3] hepatoprotective [4] and affecting adipocyte and osteoblast differentiation effects [5].

Generally, the sclerotium of *M. cocos* is peeled and processed into two products, the epidermis and the inner part. The epidermis is called Poriae Cutis in Chinese, and the inner part is still called Poria. The epidermis and inner part have similar types of compounds and different secondary metabolites contents [6], which are often used and studied separately. Both Poria and Poriae Cutis are officially recorded in the Chinese Pharmacopoeia.

The provenance of *M. cocos* is mainly distributed in the Dabie mountains area and Yunnan Province of China. Yunnan is suggested as the most satisfactory habitat because the quality of Yunnan *M. cocos* is being highly recommended all the time. Due to the large demand for it, and the knowledge of cultivation mastered easily by common people, this fungus is cultivated in large quantities. Although *M. cocos* is cultivated in Yunnan, the chemical profiles influencing biological activities may be uneven owing to various cultivation sites and different management techniques. It was reported in a previous study that the contents of pachymic acid of *M. cocos* in different regions of Yunnan varied significantly [7]. Consequently, customers are increasingly demanding some sort of proof of the geographical origin. For the sake of response to the demand, it is necessary to conduct research with respect to the authentication of geographical origin, which can also provide basic technology for the protection of specific geographical indication products [8].

To date, various analytical technologies that respond to the different chemical information of samples have been implemented for the origin identification of *M. cocos* [9,10,11]. Although these methods proved promising for the discrimination of provenance, they were separately applied. Nowadays, data fusion has been applied in the fields of food and medicine [12,13]. For example, Ni et al. [14] discovered that, based on high-performance liquid chromatography (HPLC) and Fourier transform infrared spectroscopy (FTIR) data fusion, the type and geographical origin of *Radix Paeoniae* samples could be successfully discriminated, and the fused data matrix showed a prominent result compared with the independent technique.

Data fusion strategies, which fuse the outputs of multiple complementary information to provide rich knowledge about a sample, are hoped to achieve a more accurate characterization than single pieces of information [15]. In addition to the fusion of several datum regarding one sample, the fusion of information regarding different parts was reported. For instance, Casale et al. [16] combined the near-infrared information obtained by the three parts (pileipellis, flesh and hymenium) of each individual to check the authenticity of dried *porcini* mushrooms. Studies mentioned above demonstrated that although time and effort would be taken to collect multiple complementary data, data fusion was suggested as an alternative strategy to show accurate characterization.

Infrared spectroscopy can provide the molecular functional group structure of metabolites. Liquid chromatography can characterize the exist of compounds and determinate the special compounds. Both techniques present different and complementary information, which were used for data fusion in this study. To the best of our knowledge, infrared spectroscopy was widely used for geographical classification because of the features of simplicity and rapidity [17,18]. Liquid chromatography was almost used for determining the contents of compounds [19,20]. Multiple chromatographic data fusion has been merely reported in the authentication of the geographical origin of palm oil [21], predicting antioxidant activity of *Turnera diffusa* [22], authentication of *Valeriana* species [23] as well as a comparison of *Salvia miltiorrhiza* and its variety [24]. Actually, a wealth of information was contained in the chromatographic data, and due to extensive automation, a stable and reliable result could be obtained.

In this study, two data fusion strategies including low and mid-level fusion as well as two data combinations including the fusion of complementary information regarding a single part, and the fusion of information regarding two medicinal parts from one sclerotium were performed for the geographical authentication of *M. cocos*. Liquid chromatograms at two wavelengths (242 nm and 210 nm) and FTIR spectra of two medicinal parts (Poria and Poriae Cutis) of *M. cocos* were analyzed. Contents of five triterpene acids were measured. Chromatographic data fusion, spectral data fusion as well as chromatography and spectroscopy data fusion were implemented, combined with partial least squares discriminant analysis (PLS-DA).

## 2. Results and Discussion

### 2.1. Spectral Analysis

FTIR is an auxiliary method in the structural elucidation of organic compounds, which is also employed to assess the quality attributes of a product and authenticate geographic location [17]. With the characteristics of easy operation and rapid acquisition, it was applied to the identification of cultivation location of *M. cocos*. The second derivative spectra of samples from each geographic origin were given in Figure 1, and absorption peaks were observed in the form of negative peaks. Because a 2600–1750 cm^−1^ signal was caused by ATR crystal material [25], it was discarded and did not present in the Figure.

Absorption bands at 2964 and 1704 cm^−1^ were just observed in Poriae Cutis samples. A disparity of absorption intensity exhibited in samples from different cultivation locations. Relatively high absorbance values were at around 1200–950 cm^−1^, which were mainly caused by C-O stretching vibration, C-C stretching vibration and C-OH bending vibration of polysaccharides [26,27]. Peaks located at 2964 and 2873 cm^−1^ correspond to C-H antisymmetric and symmetrical stretching vibration of methyl group respectively, while the peak at 2927 cm^−1^ is assigned to C-H antisymmetric stretching vibration of methylene. The absorption at 1452 cm^−1^ and 1373 cm^−1^ belonged to C-H antisymmetric and symmetrical bending vibration of methyl [11]. The peak at 1643 cm^−1^ was assigned to C=O antisymmetric stretching vibration, which was related to triterpenes [28]. The band at 1704 cm^−1^ was associated with C=O group of esters [29,30]. The band at 891 cm^−1^ was assigned to the bending vibration of the C=CH_2_ functional group [28]. The peak at 1259 cm^−1^ may be related to the amide III band [31]. In total, FTIR spectrum reflected comprehensive structural information of components in *M. cocos* samples, like triterpenes, polysaccharides, and so on.

### 2.2. Quantitative Analysis of Five Triterpene Acids

The content of each triterpene acid was calculated by their calibration curves and result of the validation of quantitative method was presented in Appendix A. The calibration curves of five compounds showed good linearity (R^2^ ≥ 0.99). Recovery rates calculated by the standard addition method varied from 96.32% to 106.4%. Values of relative standard deviation (RSD) of intra-day and inter-day precision were less than 1.24% and 5.68%, respectively. RSDs of repeatability did not exceed 5.95% after analyzing six solutions from the same sample in parallel. RSDs of stability were less than 0.71% after detecting a sample solution at 0, 6, 12, 17, 21 and 24 h, respectively. The method validation above indicated that the quantitative analysis was feasible. In particular, due to the obvious difference in the contents of poricoic acid A in Poria and Poriae Cutis samples, the calibration curves in two concentration ranges were prepared separately.

Contents of five triterpene acids were displayed as box-plot given in Figure 2. One-way analysis of variance was computed by SPSS 21.0 software (IBM Corporation, Armonk, NY, USA) to display the difference among eight cultivated locations. A value of *p* < 0.05 was considered significant. Poricoic acid A contents of Mengmeng were significantly different from those of Beicheng, Tuodian and Zhanhe in inner parts, and Yongping in cutis samples. Contents of dehydropachymic acid and pachymic acid in inner parts from Caodian were higher than those of other geographical origins except for Baliu. Inner parts from Baliu showed higher contents of dehydropachymic acid than those from Beicheng, Dawen and Mengmeng, and higher contents of pachymic acid than those from Tuodian, Yongping, Beicheng and Mengmeng. Inner parts from Dawen contained fairly low contents of dehydrotrametenolic acid compared with those from others with the exception of Baliu. Compared with epidermis samples from Dawen, Beicheng and Yongping showed higher contents of dehydrotumulosic acid, and Caodian and Baliu presented higher amount of pachymic acid. From the results, it was found that it was difficult to distinguish *M. cocos* samples from eight cultivation origins just in terms of contents of several target compounds. Therefore, it was necessary to take full advantage of the chromatographic fingerprint, namely, the intensity data for each retention time, to extract more information related to cultivation location.

### 2.3. Chromatographic Data Preprocessing

The chromatograms recorded at 242 nm in Appendix A were obtained by analyzing the solution from the same sample five times successively within a day and on two consecutive days. Obviously, the retention time of each peak shifted in two days, which was inconvenient for the qualitative results of chemometric analyses. Hence, all of the chromatographic data should be aligned prior to further analysis.

The correlation optimized warping algorithm proposed by Skov et al. [32] was used to correct the retention time shifts among samples. The chromatogram that was most similar to all others was selected to be the reference chromatogram for alignment. The global search space was set to a combination of segment length from 10 to 200 and a slack size from 1 to 20 according to the observed peak widths and shifts on the chromatograms. Then the optimal combination of segment length slack size was automatically selected according to the criterion of well alignment while at the same time considering the preservation in peak shape and area. The theory for the algorithms with respect to the automated alignment of chromatographic data can be consulted in [32].

As a result, suitable combinations of segment length and slack size were achieved for chromatographic data at 242 nm of Poria (197 and 11), 210 nm of Poria (105 and 16), 242 nm of Poriae Cutis (105 and 11) and 210 nm of Poriae Cutis (198 and 16), respectively. Figure 3 presented the aligned *M. cocos* chromatographic fingerprints using these warping parameters, which displayed that the retention time shifts were properly corrected. What’s more, it was observed that chromatograms of the same medicinal part recorded at 242 nm and 210 nm showed complementary information, i.e., some peaks obviously presented in liquid chromatograms at 242 nm (LC_242_) and some compounds just displayed in liquid chromatograms at 210 nm (LC_210_). Further, chromatograms of two parts were appreciably different. In other words, multiple chromatographic profiles presented abundant chemical information of *M. cocos* that probably facilitated to confirm cultivation areas.

The chromatographic data of one Poria sample and one Poriae Cutis sample could be represented as 7201 and 7801 data points, respectively. In order to save the time for calculation, the number of data points in the retention time dimension of the matrix was reduced by taking one in every three points without affecting the chromatographic features. Therefore, 2401 and 2601 data points were obtained after reducing data, respectively. It was proved that this method was feasible by comparing the PLS-DA results since reducing data had little influence on identifying different groups (Appendix A). Additionally, the first 11 min data in the chromatogram that mainly comprised unseparated peaks and baseline shift (Figure 3), which were discarded to obtain representative fingerprints and accurate results. In this way, the final data points were 1960 and 2160, respectively.

### 2.4. PLS-DA Using Chromatograms and FTIR Spectra

Partial least squares discriminant analysis is a widely-used linear classification method [33,34,35,36]. The selection of the optimal number of latent variables was an essential question for PLS-DA model, which was implemented on the basis of 7-fold cross validation procedure in present study. Unit variance scaling, which could give all variables of the same or different measurements equal importance, was performed by default when developing each PLS-DA model. The parameters of classification models were shown in Table 1 and Appendix A in detail.

Based on the preprocessing of chromatograms and FTIR spectra, a model of PLS-DA was established using the single dataset (Table 1 and Appendix A). The LC_210_ dataset of Poriae Cutis samples did not build model successfully, so results of classification were not listed. FTIR and LC_242_ datasets showed better performance with higher accuracy not only in calibration set but in validation set than LC_210_ dataset. The sensitivity values of class 2 and class 8 in the validation set were 1 for Poria LC_242_ model and were smaller values for the Poria FTIR model, which indicated that LC_242_ model had stronger ability to correctly recognizing samples of class 2 and class 8. While the sensitivity of class 1 and 7 in calibration set was 0.8571 for Poria LC_242_ model smaller than that of Poria FTIR model, indicating that FTIR model had stronger ability to correctly recognizing samples of class 1 and class 7. Moreover, LC models of Poriae Cutis samples presented poorer results than those of Poria samples, which reflected the difference of two medicinal parts of *M. cocos*.

Variable importance for the projection (VIP) plot [37] was used for assessing the significance of variable, and that the VIP score of retention time was greater than one means the compound separated at the time was important on distinguishing different cultivation origins. As an example of the Poria LC_242_ model, there were lots of variables whose VIP were higher than one including the corresponding retention time of poricoic acid A and dehydrotrametenolic acid (Figure 4). It indicated that the potential of the chromatographic fingerprint from the aspect of origin identification was greater than that of the contents of several compounds. However, all single technique models did not achieve a perfect performance, so it was necessary to carry out the data fusion strategy that was expected to enhance the classification and prediction ability of the model.

### 2.5. Low-Level Data Fusion

#### 2.5.1. PLS-DA of Poria

Figure 5 was the workflow of geographical authentication using data fusion, which was helpful to understand how data was combined. As shown in Table 1, accuracy rates of low-level data fusion datasets about Poria samples were 100% and higher than those of single technique models except for the model using LC_242-210_ data, which implied that these models had strong classification performance. The highest R^2^(cum) (0.9599) and Q^2^(cum) (0.7917) were observed in FTIR-LC_242_ model, indicating a high goodness of fit for the established model in the data and good predictive ability. Therefore, the combination of FTIR and LC_242_ datasets was deemed a suitable strategy, and the fusion of three datasets was unnecessary and verbose. Furthermore, compared with the LC_242-210_ model, the accuracy of FTIR-LC_210_ model was higher both in calibration and validation sets. It could be interpreted that FTIR dataset provided more helpful information to identify eight geographical origins than LC_242_ dataset in data fusion model of Poria samples. By analogy, it was found that FTIR data showed more contribution for origin discrimination than LC_210_ data when compared LC_242-210_ model with FTIR-LC_242_ model.

#### 2.5.2. PLS-DA of Poriae Cutis

The accuracy of FTIR-LC_242_ and FTIR-LC_242-210_ models was 100%, which was greater than that of the models using the independent technique. It indicated the effectiveness of low-level data fusion. The similar Q^2^(cum) of FTIR-LC_242_ and FTIR-LC_242-210_ models was observed. Accordingly, FTIR-LC_242_ was considered as a preferred combination, and the fusion of three datasets was superfluous. Furthermore, the Q^2^(cum) values of low-level fusion models about Poriae Cutis samples (≤ 0.7032) were less than those of corresponding models about Poria samples (> 0.75), indicating that Poria samples were more suitable for origins identification than Poriae Cutis species. In the developing LC_242-210_ and FTIR-LC_210_ low-level models, latent variables could not be calculated so the models were not successfully built. It was in consistent with the state that epidermis LC_210_ dataset did not built PLS-DA model, which was probably attributed by a lot of irrelevant classification information included in LC_210_ dataset of epidermis.

#### 2.5.3. PLS-DA of Combination Data of Two Medicinal Parts

Both FTIR and LC_242_ datasets of two parts samples showed better performance than LC_210_ dataset, which was in accordance with the results of single technique mentioned above. Compared with the single spectrum or chromatogram, data fusion of two medicinal parts proved more advantageous with greater sensitivity, specificity and efficiency. Therein, the FTIR fusion model of two part samples presented the best prediction performance from the Q^2^(cum) point of view. What’s more, compared with FTIR-LC_242_ model of Poria samples, the Q^2^(cum) of LC_242_ fusion model of two parts was smaller. It could be interpreted that Poria FTIR dataset provided more helpful information to predict different geographical origins than Poriae Cutis LC_242_ dataset in data fusion model. By analogy, it was found that the contribution of FTIR dataset was always more than that of LC_242_ and LC_210_ datasets in low-level data fusion. The low-level data fusion strategy has achieved a good classification result, but the mid-level data fusion could spend less computation time compared to the low level. Therefore, mid-level fusion was performed.

### 2.6. Mid-Level Data Fusion

#### 2.6.1. The Extraction of Feature Variables

Mid-level fusion needed to first extract relevant features from each dataset independently and then concatenated them into a new matrix employed for origins identification. Principal component analysis (PCA) is a dimension reduction technique that creates a small number of new variables called principal components (PCs) from a large number of original variables, which would be applied to extract features. These PCs almost retain the same information as the original variables [38]. The optimal number of PCs was determined by 7-fold cross-validation procedure. The results of feature extraction were shown in Appendix A. As an example of LC_210_ dataset of Poria samples, the first thirteen PCs were extracted, which account for 90.92% of the information concerning the original variables. Then the scores of the thirteen PCs were used for data fusion.

#### 2.6.2. PLS-DA of Poria

In agreement with the results of low-level data fusion, the accuracy rates of FTIR-LC_242_ and FTIR-LC_242-210_ of Poria samples were 100% not only in calibration set but in validation set. And they had stronger recognition performance with higher sensitivity, specificity, efficiency than corresponding single dataset. Nonetheless, all Q^2^(cum) values of mid-level data fusion models of Poria samples were less than those of low-level data fusion models, indicating that low-level fusion presented stronger prediction ability than mid-level fusion according to cross validation.

As always, The LC_242-210_ fusion model did not build successfully. The fusion of LC_242_ and LC_210_ could not gain satisfactory discrimination and even could not construct the model, and it was likely caused by the similar chemical information provided by both chromatograms. Although they presented different peak shapes, there were many common chromatographic peaks that did not provide complementary and useful information.

#### 2.6.3. PLS-DA of Poriae Cutis

LC_242-210_ model that was not built successfully in low-level fusion finished construction in mid-level fusion. The fact indicated the significance of mid-level data fusion and might be due to the feature extraction. The accuracy rates of FTIR-LC_210_ and FTIR-LC_242-210_ models were equal, but the detail of incorrect identification was different from sensitivity and specificity points of view. Further analysis showed that one sample belonging to Tuodian was judged as the sample from Baliu in FTIR-LC_210_ model and Mengmeng in FTIR-LC_242-210_ model by mistake, respectively. FTIR-LC_242_ and FTIR-LC_242-210_ mid-level fusion models of Poriae Cutis samples presented poorer results than those of Poria samples as well as low-level data fusion models and FTIR model of epidermis samples.

#### 2.6.4. PLS-DA of Combination Data of Two Medicinal Parts

Both FTIR data fusion and LC_242_ data fusion of two medicinal parts had stronger recognition ability when compared to the LC_210_ combination. Both LC_242_ and LC_210_ of two medicinal parts improved performance of single LC_242_ and LC_210_ models. However, the result of FTIR was the opposite. Compared to low-level data fusion, the identification ability of mid-level data fusion did not show any obvious advantage. This might be due to the limitation of our method of feature extraction. In terms of FTIR datasets, only more than 73.29% original information (Appendix A) was extracted.

To validate the performance of the PLS-DA model, a 30-iteration permutation test was performed. As shown in Appendix A that one of permutations plots for Poria LC_242-210_ model, all permutated Q^2^ and R^2^ values (bottom left) were lower than the corresponding original values (top right). It indicated that the PLS-DA model was considered as an appropriate model without randomness and overfitting. The results showed that all the PLS-DA models were not overfitting.

## 3. Materials and Methods

### 3.1. Reagents, Solvents and Standard References

Dehydrotumulosic acid (purity ≥ 96%) was supplied by ANPEL Laboratory Technologies Inc. (Shanghai, China). Dehydropachymic acid, pachymic acid, poricoic acid A and dehydrotrametenolic acid (purity ≥ 98%) were purchased from Beijing Keliang Technology Co., Ltd. (Beijing, China). HPLC grade acetonitrile and formic acid were purchased from Thermo Fisher Scientific (Fair Lawn, NJ, USA) and Dikma Technologies (Lake Forest, CA, USA), respectively. Purified water was purchased from Guangzhou Watsons Food & Beverage Co., Ltd. (Guangzhou, China). Other chemicals and reagents were analytical grade.

### 3.2. Samples

Seventy-eight intact cultivated *M. cocos* sclerotia (Figure 6) from eight geographical origins of Yunnan Province, China were collected and identified by Prof. Yuanzhong Wang (Institute of Medicinal Plant, Yunnan Academy of Agricultural Sciences, Kunming, China). Voucher specimens (FL20160217) were deposited in the herbarium of Institute of Medicinal Plant, Yunnan Academy of Agricultural Sciences. After digging sclerotium up, the soil was brushed away. Fresh *M. cocos* sclerotium was air-dried in the shade and then peeled. Both the epidermis and inner part of the dried sclerotium, i.e., Poria and Poriae Cutis, were powdered to a homogeneous size using pulverizer and sieved through No. 60 mesh sieve. The powder was stored in the airproof, dry and dark condition prior to analysis. Detailed information of samples was summarized in Table 2.

### 3.3. FTIR Spectra Acquisition

A Fourier transform infrared spectrometer from Perkin Elmer equipped with an attenuated total reflectance (ATR) sampling accessory with a diamond focusing element was employed for FTIR spectroscopy measurement. The sample powder was pressed under a consistent pressure with a pressure tower when collecting spectral. FTIR spectrum of each sample was scanned 16 times successively with a resolution of 4 cm^−1^ in the range of 4000–650 cm^−1^. After the measurement of one sample was finished, the surface of ATR crystal and the apex of pressure tower were cleaned for the next sample detection. All spectra were background corrected utilizing air spectrum. The laboratory environment was maintained a constant temperature (25 °C) and humidity (30%).

### 3.4. Chromatographic Analysis

Sample powder was weighed accurately to 0.5 g and extracted with 2.0 mL of methanol by an ultrasound-assisted method for 40 min at ambient temperature. The extract solution was filtered using a 0.22 μm membrane filter. The filtrate was loaded into the auto-sampler vial and stored at 4 °C before injecting into the chromatographic system for analysis.

Analyses of all 156 samples (including Poria and Poriae Cutis) were implemented using a Shimadzu ultra-fast liquid chromatography system equipped with a UV detector, binary gradient pumps, a degasser, an auto sampler and a column oven. The chromatographic separation was achieved using an Inertsil ODS-HL HP column (3.0 × 150 mm, 3 μm particle size) operated at 40 °C. The mobile phase consisted of acetonitrile (A) and 0.05% formic acid (B). Before use, the mobile phase constituents were degassed and filtered through a 0.2 μm filter. The gradient elution sequence was conducted as follows: 0–25 min, 40% A; 25–52 min, 40–69% A; 52–56 min, 69–72% A; 56–58 min, 72–78% A; 58–58.01 min, 78–90% A; and 58.01–60 min, remaining at 90% A (eluting to 65 min for Poriae Cutis samples). Each run was followed by an equilibration period of 3 min with initial conditions (40% A and 60% B). The flow rate was kept at 0.4 mL·min^−1^ and the injection volume was 7 μL. Detective wavelengths were set at 242 nm and 210 nm.

### 3.5. Method Validation

The developed UFLC method was validated in terms of precision, stability, repeatability, accuracy and linearity under the above chromatographic condition.

A mixed standard solution was determined six times successively within a day and on three consecutive days for evaluating intra- and inter-day precision. For the stability test, the extract of a sample was analyzed at 0, 6, 12, 17, 21 and 24 h, respectively. Six sample solutions prepared individually from the same sample were analyzed in parallel for evaluating the repeatability. The recovery test was performed to evaluate the accuracy by adding reference compounds of three different amounts (low, middle, and high) to the sample with known concentration accurately. The following equation was used to calculate recovery rate: Recovery rate (%) = [(measured amount − original amount)/spiked amount] × 100%.

The standard solutions of five compounds for constructing calibration curves were prepared by diluting the stock solutions with methanol individually. The ranges of concentration in the linearity study were 5.00–999 μg·mL^−1^ (dehydrotumulosic acid), 0.22–6730 μg·mL^−1^ (poricoic acid A), 2.4–480 μg·mL^−1^ (dehydropachymic acid), 10.3–1240 μg·mL^−1^ (pachymic acid) and 0.49–2450 μg·mL^−1^ (dehydrotrametenolic acid). Due to the obvious difference in contents of poricoic acid A of Poria and Poriae Cutis samples, two concentration ranges of 0.22–1121.95 μg·mL^−1^ (Poria) and 0.22–6730 μg·mL^−1^ (Poriae Cutis) were prepared. More than seven levels (in arithmetic progression) of every concentration range were guaranteed. The limit of detection (LOD) and limit of quantification (LOQ) were determined by diluting continuously standard solution until the signal-to-noise ratios (S/N) reached about 3 and 10, respectively.

### 3.6. Preprocessing of Chromatograms and Spectra

The correlation optimized warping algorithm was applied to correct the retention time shifts of chromatogram using MATLAB software (MathWorks, R2017a, Natick, MA, USA). Then the corrected chromatographic data was reduced by taking one in every three points without affecting the chromatographic features to save computation time, which was inspired by the ‘data binning’ of Lucio-Gutiérrez et al. [22,23]. The first 11 minutes of data that mainly comprised unseparated peaks and baseline shift were discarded.

Raw FTIR spectra were subjected to advanced ATR correction to reduce the impact of skewing of band intensity using OMNIC 9.7.7 software (Thermo Fisher Scientific). Due to the fact that spectra contained hidden and overlapped absorption peaks, second derivative was used for highlighting slight differences employing SIMCA-P^+^ 13.0 software (Umetrics, Umeå, Sweden). Derivative spectra were calculated with a Savitzky–Golay filter using a second-order polynomial and a 15-point window. The band of 2600–1750 cm^−1^ was associated to diamond crystal in ATR accessory, of which data were excluded prior to chemometrics analysis. These pre-processed data were used to data fusion and PLS-DA.

### 3.7. Multiple Chromatograms and Spectra Data Fusion

According to the source of data, there were two kinds of data fusion techniques, including the fusion of multiple complementary pieces of information about a single part and the fusion of information about two parts from one sclerotium. For instance, data matrices of LC-Poria and FTIR-Poria could be fused into a new dataset, and data matrices of FTIR-Poria and FTIR-epidermis could be fused into a dataset. It was important to note that information must correspond in the process of data fusion, namely, the LC and FTIR data of the same Poria sample must correspond, or the FTIR data of inner parts and epidermis from the same sclerotium should correspond.

The data fusion could be classified into three levels in light of the combination of data: low level, mid-level and high level. Low and mid-level fusion has been widely used, and was applied to the identification of geographical origin of *M. cocos*. The scheme of low and mid-level data fusion approaches is shown in Figure 7. In the low-level fusion, pre-processed different datasets were straightforward concatenated into a matrix, and the number of variables was equal to the sum of number of original variables. For the mid-level fusion, the scores obtained independently from different data by PCA were concatenated into a dataset applied for provenance traceability, and the number of variables of the dataset was significantly less than that of original variables. Compared with low level, mid-level data fusion could save more time on the operation. Specific types of the data fusion in this study were shown in Table 1.

### 3.8. Evaluation of Model Performance

The calibration and validation sets were selected for assessing the quality of model. The calibration set was used to construct a model that was performed 7-fold cross validation for internal validation, and the validation set was used to externally estimate the practicability of model. To avoid the influence of randomness caused by random sampling, and to obtain a representative calibration set from a pool of samples, the Kennard-Stone algorithm [39] was performed to systematically divide dataset of 78 samples into calibration (52) and validation (26) sets using MATLAB R2017a software (MathWorks).

The performance of discrimination model could be evaluated by sensitivity, specificity and efficiency [40]. The three parameters are dependent on these values: true positive (TP), false positive (FP), true negative (TN) and false negative (FN). TP and TN represent the correctly identified samples in target positive and negative classes, respectively. By analogy, FP and FN represent the incorrectly identified samples in positive and negative classes, respectively.
(1)$$\mathrm{Sensitivity}\;=\;\frac{\mathrm{TP}}{\mathrm{TP}\;+\;\mathrm{FN}}$$
(2)$$\mathrm{Specificity}\;=\;\frac{\mathrm{TN}}{\mathrm{TN}\;+\;\mathrm{FP}}$$
(3)$$\mathrm{Efficiency}\;=\;\sqrt{\mathrm{sensitivity}\;\times \;\mathrm{specificity}}$$

Therein, sensitivity shows the ability to correctly recognize samples belonging to the target class while specificity reflects the model ability to reject samples belonging to all other classes. The measure combining the sensitivity and specificity value is called efficiency.

In addition, the accuracy rate of calibration set, the accuracy rate of validation set, R^2^(cum) and Q^2^(cum) were also employed for assessing the classification performance. Accuracy was obtained by calculating the proportion of correctly classified samples in the total amount of calibration set (or validation set) samples. R^2^ is calculated by following equation: R^2^ = 1 − RSS/SSX, where RSS is the residual sum of squares of calculated and measured values, and SSX is the total sum of squares after mean centralization [41]. R^2^(cum) represents the percentage of explained variance for a defined number of latent variables, indicating how well the model fits the data. Q^2^(cum) represents the cross-validated cumulative R2, suggesting how well the model predicts new data. The higher values of these parameters (close to 1 or 100%), the better performance of model.

## 4. Conclusions

In order to establish an effective method for geographical authentication of *M. cocos*, two data fusion strategies, including low and mid-level fusion, as well as two data combinations, including the fusion of complementary information regarding a single part and the fusion of information about two parts from one sclerotium were compared. FTIR, LC_242_ and LC_210_ were used to characterize the epidermis and inner part of *M. cocos* sclerotium from different places individually and jointly. The results showed that, chromatographic fingerprint was more suitable than content data of five triterpene acids for origin identification. In the fusion of complementary information about single part, good classification performance was achieved obtained by merging LC_242_ chromatograms and FTIR spectra in low-level fusion way. In the fusion of information about two parts from one sclerotium, the predictive ability of the FTIR low-level fusion model of two parts was the most satisfactory, and all analyzed samples were classified correctly.

In most cases, FTIR proved to be more efficient than LC_242_ and LC_210_, not only in a single data source but in data fusion. Mid-level data fusion was slightly worse than low-level data fusion. The performance of low-level data fusion models was superior to single technique models. Moreover, Poria samples were more suitable for origin identification than Poriae Cutis samples. On the basis of effective and comprehensive fingerprint information, the low-level data fusion strategy could be used for the discrimination of *M. cocos* samples from different origins with the aid of appropriate mathematical algorithms.

## Figures and Tables

**Figure 1 molecules-24-01320-f001:**
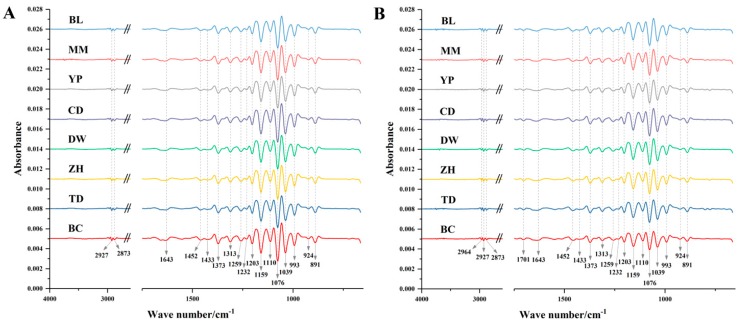
Second derivative spectra of Poria (**A**) and Poriae Cutis (**B**) samples from eight geographic origins.

**Figure 2 molecules-24-01320-f002:**
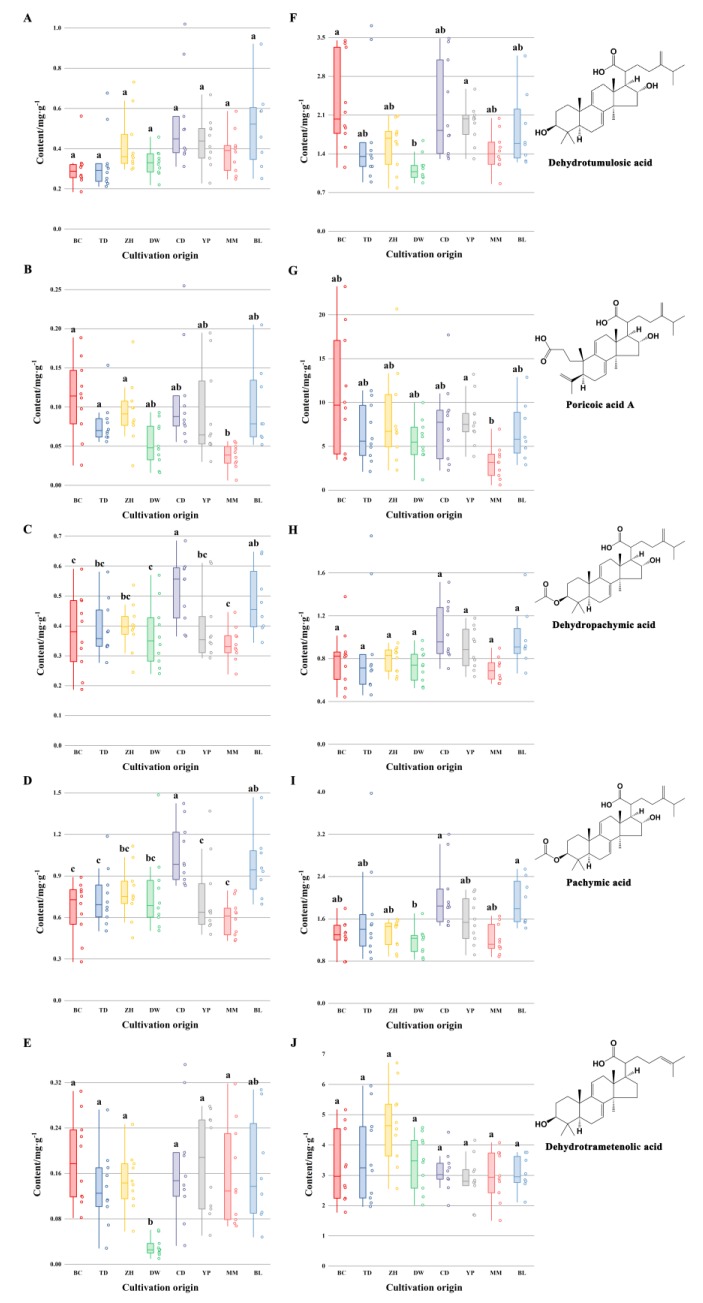
Box-plots of contents of five triterpene acids of Poria (**A**–**E**) and Poriae Cutis (**F**–**J**) samples from eight geographical origins. Note: Different letters show significant difference (*p* < 0.05).

**Figure 3 molecules-24-01320-f003:**
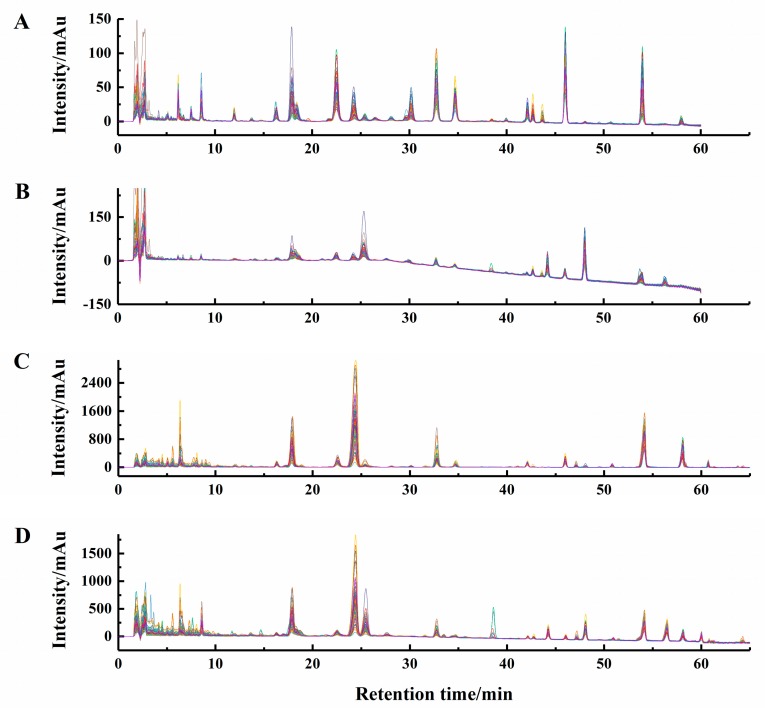
Chromatograms of Poria (**A**,**B**) and Poriae Cutis (**C**,**D**) recorded at 242 (**A**,**C**) and 210 nm (**B**,**D**) after the transformation of correlation optimized warping.

**Figure 4 molecules-24-01320-f004:**
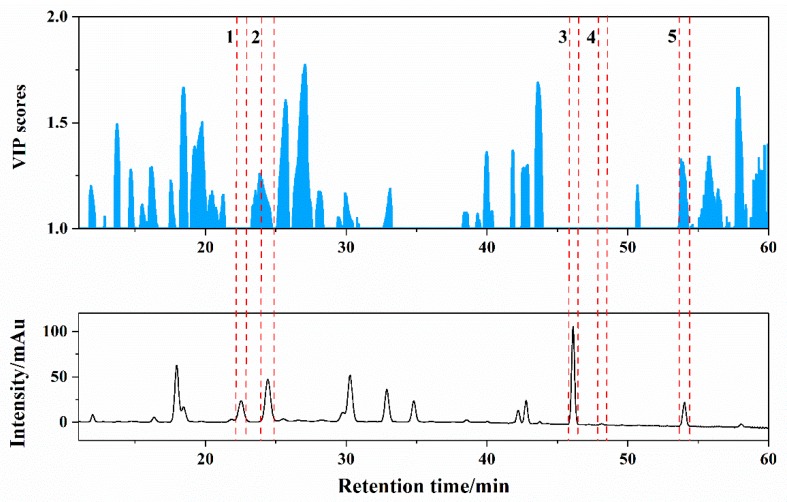
VIP scores of PLS-DA using LC_242_ chromatogram data of Poria samples. Note: 1, dehydrotumulosic acid; 2, poricoic acid A; 3, dehydropachymic acid; 4, pachymic acid; 5, dehydrotrametenolic acid.

**Figure 5 molecules-24-01320-f005:**
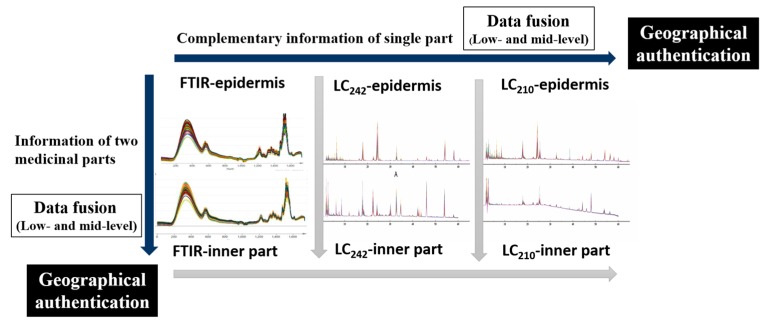
The workflow of geographical authentication using data fusion.

**Figure 6 molecules-24-01320-f006:**
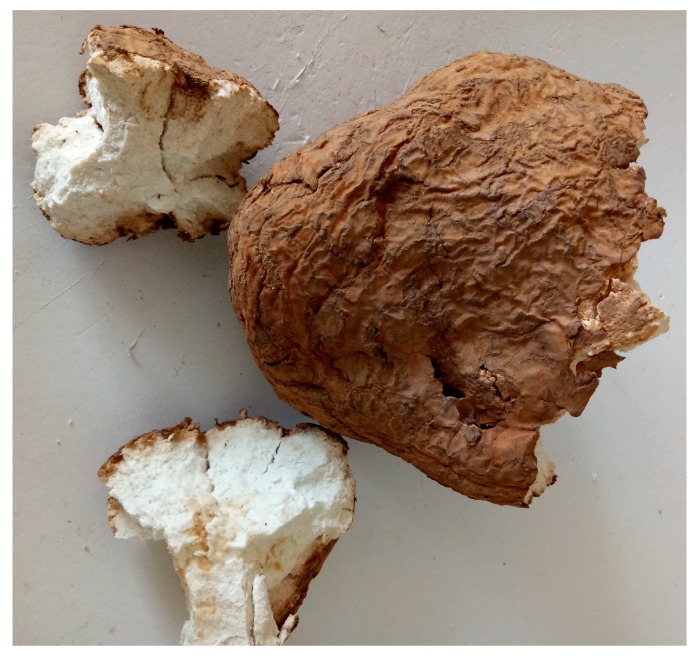
Dried sclerotium of *M. cocos*.

**Figure 7 molecules-24-01320-f007:**
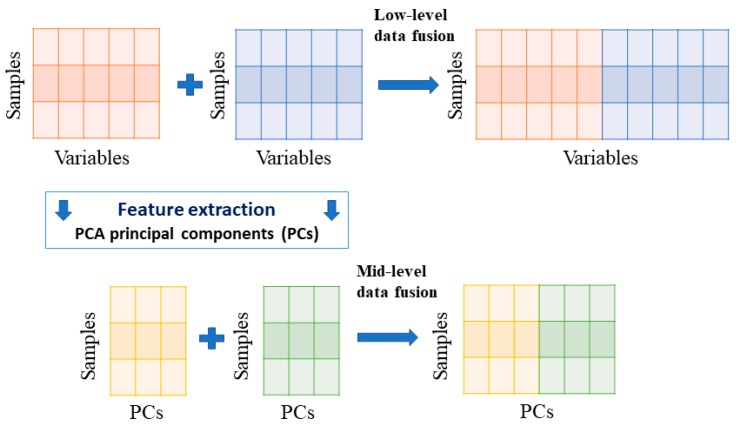
The scheme of the data fusion approaches.

**Table 1 molecules-24-01320-t001:** The major parameters of PLS-DA model.

Fusion Approach	Data Matrix		Calibration Set			Validation Set
			R^2^(cum)	Q^2^(cum)	Accuracy	Accuracy
single technique	Poria	FTIR	0.8883	0.7268	100%	92.31%
		LC_242_	0.6634	0.5277	96.15%	100%
		LC_210_	0.5174	0.4012	90.38%	76.92%
	Poria Cutis	FTIR	0.9292	0.6981	100%	96.15%
		LC_242_	0.2874	0.2204	65.38%	34.62%
low-level data fusion	Poria	FTIR-LC_242_	0.9599	0.7917	100%	100%
		FTIR-LC_210_	0.9468	0.7663	100%	100%
		LC_242-210_	0.8097	0.6547	98.08%	92.31%
		FTIR-LC_242-210_	0.8823	0.7566	100%	100%
	Poria Cutis	FTIR-LC_242_	0.9016	0.7032	100%	100%
		FTIR-LC_242-210_	0.905	0.698	100%	100%
	combination data of two medicinal parts	FTIR	0.9548	0.8064	100%	100%
		LC_242_	0.8147	0.6495	100%	100%
		LC_210_	0.6489	0.4806	94.23%	88.46%
mid-level data fusion	Poria	FTIR-LC_242_	0.8266	0.5745	100%	100%
		FTIR-LC_210_	0.7453	0.5053	96.15%	96.15%
		FTIR-LC_242-210_	0.8286	0.5882	100%	100%
	Poria Cutis	FTIR-LC_242_	0.7386	0.5493	100%	92.31%
		FTIR-LC_210_	0.7518	0.4991	100%	96.15%
		LC_242-210_	0.4617	0.228	76.92%	73.08%
		FTIR-LC_242-210_	0.7607	0.5558	100%	96.15%
	combination data of two medicinal parts	FTIR	0.7564	0.5982	98.08%	88.46%
		LC_242_	0.7761	0.4973	98.08%	100%
		LC_210_	0.676	0.3756	96.15%	88.46%

**Table 2 molecules-24-01320-t002:** The information of *M. cocos* samples.

Class	Location	Abbreviation	Elevation (m)	Latitude (°N)	Longitude (°E)	Parts	Sample Size
1	Beicheng Town, Hongta, Yuxi	BC	1720	24.4319	102.5182	inner part	10
						epidermis	10
2	Tuodian Town, Shuangbai, Chuxiong	TD	2062	24.6912	101.6493	inner part	10
						epidermis	10
3	Zhanhe Town, Ninglang, Lijiang	ZH	2560	26.8832	100.9275	inner part	10
						epidermis	10
4	Dawen Town, Shuangjiang, Lincang	DW	1438	23.3487	100.0047	inner part	10
						epidermis	10
5	Caodian Town, Yunlong, Dali	CD	2066	25.6360	99.1320	inner part	10
						epidermis	10
6	Yongping Town, Jinggu, Pu’er	YP	1077	23.4204	100.4044	inner part	10
						epidermis	10
7	Mengmeng Town, Shuangjiang, Lincang	MM	1052	23.4779	99.8378	inner part	10
						epidermis	10
8	Baliu Town, Mojiang, Pu’er	BL	1979	23.0676	101.9765	inner part	8
						epidermis	8

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
