# Peer review of "Geographical Authentication of Macrohyporia cocos by a Data Fusion Method Combining Ultra-Fast Liquid Chromatography and Fourier Transform Infrared Spectroscopy"

_molecules, 2019, doi:10.3390/molecules24071320_

Round 1

Reviewer 1 Report

The topic of the research is reasonable and its relevancy in the literature is well described and contextualized. Nevertheless, in my opinion, few modifications are needed, in order to clarify nomenclature and procedures, with the aim of increasing the general readability of the paper.

Major comments:

Section 3.7

In Section 3.7 the authors properly describe how the evaluation of the models performances is done. Nevertheless, I have few question about the figures of merit used to evaluate these performances.

Is the R2 the coefficient of determination? Please, state it and describe how it is calculated. Additionally, its description (lines 383-384, page 12) is not very clear to me. What do you mean with interpretation ability? How to interpret classification models is a complex topic, still widely discussed; for the sake of clearness, in order to help the reader, I would suggest the authors to re-write this sentence.   

Section 3.5

In Section 3.5 the authors describe pretreatments applied on the different data sets.
This section is well organized and several details are provided to the reader. I personally agree with the authors second derivative could be a good solution for FTIR spectra, but, as a reader, I would ask myself why, among all the possible pretreatments developed for spectroscopic data, this specific preprocessing approach has been used. Have the authors tested somehow (in a cross-validation procedure, for instance) if this approach is more suitable than others? In case they have, please mention it and discuss your choice.

Section 2.4, Table 1

A detailed description of the results is reported in Table 1; this, together with tables reported in the supplementary material, provide a complete overview of the results, which allow the comprehension of the study and supports its feasibility.  

The authors properly display classification results obtained when PLS-DA is calculated on individual data blocks or by means of low- and mid-level data fusion approaches.

From the table is clear the low-level strategy is definitely the most suitable approach to handle the data set under study, in fact, as expected, it provided comparable or better results than PLS-DA on FTIR. From the table, it is also clear that the mid-level approach is less performant (in terms of predictions) than the low-level approach. In my opinion, this is not completely unexpected, but I think it would be better to provide to the reader a tentative reason/some considerations about it.

Additionally, for the sake of completeness, I think it would be appropriate to mention why the mid-level approach is reported, despite it provides less accurate results than the low-level.

References

The topic of the paper is well described and the rationale behind the study is supported by several references. Nevertheless, I would suggest the authors to include some other references, for different reasons, described below.

Introduction

As above-mentioned, the introduction is well-organized, and the feasibility of developing an authentication procedure for Machrohyporia cocos is well described and supported. The peculiarities of this fungi are presented and several references about its characteristics are reported; nevertheless, in my opinion, in the final part of the introduction, where the chemometric tools are described, some references are missing. So far, the choice of some chemometric approaches is not well supported by the cited literature, and I think it is appropriate to address the reader to some of the basic literature on this regard.

For instance, discussing the relevance of tracing fungi, I would address the reader to some literature which would provide an overview on classification in a similar context.  

For instance, at line 58 page 2, I would add the following reference:

Biancolillo, A.; Marini, F. Chapter Four - Chemometrics Applied to Plant Spectral Analysis. In Vibrational Spectroscopy for Plant Varieties and Cultivars Characterization, Comprehensive Analytical Chemistry, 1st ed.; Lopes, J., Sousa, C. Eds.; Elsevier: Amsterdam, Nederlands, 2018; Volume 80, pp. 69-104.

Concerning other studies on medicinal herbs (analysed by chromatography), I am not completely aware of this specific topic, but, if possible, I would try to widen the state of the art (lines 68-70 pag2). 

Concerning data fusion applied for the analysis of mushrooms, I would suggest you to provide a wider description of the state of the art, for instance, I am aware of another study about it, which could be mentioned:

-Yao, S.; Li, T.; Liu, H.G.; Li, J. Q.; Wang Y. Z. Geographic Characterization of Leccinum rugosiceps by Ultraviolet and Infrared Spectral Fusion. Anal Lett 2017, 50, 2257–2269.

And maybe some others are available in the literature; please check and in case enlarge the references including other studies on the same topic.

Section 2.4

The reference reported ([29]) is for sure very good. Nevertheless, mentioning a method, it would be nice to give reference also to the first author(s) who discussed it. Consequently, I would suggest you to add at least the following references to PLS-DA:

- Sjöström, M.; Wold, S.; Söderström, B. PLS discriminant plots. In Pattern recognition in practice; Gelsema, E.S., Kanal L.N. Eds.; Elsevier: Amsterdam, Netherlands, 1986; pp. 461-470.

-Barker, M.; Rayens, W. Partial least squares for discrimination. J Chemometr 2003, 17, 166-173.

-Ståle, L.; Wold, S. Partial least squares analysis with cross-validation for the two-class problem: a Monte Carlo study. J Chemometr 1987, 1, 185-196.

The reference for VIP is missing, I would add at least the following one, in order to address the reader to the theory:

- Wold, S.; Johansson, E.; Cocchi. M. PLS: partial least squares projections to latent structures. In 3D QSAR in Drug Design: Theory, Methods and Applications, H. Kubinyi Ed.; KLUWER ESCOM Science Publisher: Leiden, The Netherlands, 1993, pp. 523-550.

Minor comments:

Line 117- page 3

The first time (in the manuscript) an acronym is mentioned in the text, please, report it together with the extended name. For instance, at line 117 of page 3 RSD is used; if it is not already present somewhere else into the manuscript, please provide the extended name.

Line 254 – page 9

Same comment as above. If I am not wrong, PCA is used for the first time at this point of the manuscript, but it the name is not reported in the extended form.

Author Response

Responds to reviewer1 comments

Point 1: In Section 3.7 the authors properly describe how the evaluation of the model’s performances is done. Nevertheless, I have few questions about the figures of merit used to evaluate these performances.

Is the R2 the coefficient of determination? Please, state it and describe how it is calculated. Additionally, its description (lines 383-384, page 12) is not very clear to me. What do you mean with interpretation ability? How to interpret classification models is a complex topic, still widely discussed; for the sake of clearness, in order to help the reader, I would suggest the authors to re-write this sentence.

Response 1: We have rewritten as “R2 is calculated by following equation: R2 = 1-RSS/SSX, where RSS is the residual sum of squares of calculated and measured values, and SSX is the total variance of model data after mean centralization [41]. R2(cum) represents the percentage of explained variance for a defined number of latent variables, indicating how well the model fits the data” in the revised manuscript.

[41] A, J. Y. Analysis of metabolomic data: principal component analysis. Chin J Clin Pharmacol Ther 2010, 15, 481-489.

Point 2: In Section 3.5 the authors describe pretreatments applied on the different data sets. This section is well organized, and several details are provided to the reader. I personally agree with the authors second derivative could be a good solution for FTIR spectra, but, as a reader, I would ask myself why, among all the possible pretreatments developed for spectroscopic data, this specific preprocessing approach has been used. Have the authors tested somehow (in a cross-validation procedure, for instance) if this approach is more suitable than others? In case they have, please mention it and discuss your choice.

Response 2: Thanks for your advice. We chose the pretreatment of second derivative depending on the experience of our research group [1,2]. In this study, we just tested a part of FTIR matrix, so there was not a systematic way and result to compare pretreatments. We have kept this problem in mind.

[1] Pei, Y.; Wu, L.; Zhang, Q.; Wang, Y. Geographical traceability of cultivated Paris polyphylla var. yunnanensis using ATR-FTMIR spectroscopy with three mathematical algorithms. Anal Methods-Uk 2019, 11, 113-122.

[2] Li, Y.; Zhang, J.; Wang, Y. FT-MIR and NIR spectral data fusion: a synergetic strategy for the geographical traceability of Panax notoginseng. Anal Bioanal Chem 2018, 410, 91-103.

Point 3: From the table is clear the low-level strategy is definitely the most suitable approach to handle the data set under study, in fact, as expected, it provided comparable or better results than PLS-DA on FTIR. From the table, it is also clear that the mid-level approach is less performant (in terms of predictions) than the low-level approach. In my opinion, this is not completely unexpected, but I think it would be better to provide to the reader a tentative reason/some consideration about it.

Response 3: The possible reason why the mid-level approach is less performant than the low-level approach is the limitation of our method of features extraction. In terms of FTIR datasets, only more than 73.29% original information (Table S7) was extracted. We have supplemented it in the revised manuscript.

Point 4: Additionally, for the sake of completeness, I think it would be appropriate to mention why the mid-level approach is reported, despite it provides less accurate results than the low-level.

Response 4: We completely agree the suggestion of reviewer. Mid-level data fusion spends less computation time compared with that of low-level fusion. So mid-level fusion is performed. This content has been added in the revised manuscript.

Point 5: References

The topic of the paper is well described and the rationale behind the study is supported by several references. Nevertheless, I would suggest the authors to include some other references, for different reasons, described below.

Introduction

As above-mentioned, the introduction is well-organized, and the feasibility of developing an authentication procedure for Machrohyporia cocos is well described and supported. The peculiarities of this fungi are presented and several references about its characteristics are reported; nevertheless, in my opinion, in the final part of the introduction, where the chemometric tools are described, some references are missing. So far, the choice of some chemometric approaches is not well supported by the cited literature, and I think it is appropriate to address the reader to some of the basic literature on this regard.

For instance, discussing the relevance of tracing fungi, I would address the reader to some literature which would provide an overview on classification in a similar context.

For instance, at line 58 page 2, I would add the following reference:

Biancolillo, A.; Marini, F. Chapter Four - Chemometrics Applied to Plant Spectral Analysis. In Vibrational Spectroscopy for Plant Varieties and Cultivars Characterization, Comprehensive Analytical Chemistry, 1st ed.; Lopes, J., Sousa, C. Eds.; Elsevier: Amsterdam, Nederlands, 2018; Volume 80, pp. 69-104.

Concerning other studies on medicinal herbs (analysed by chromatography), I am not completely aware of this specific topic, but, if possible, I would try to widen the state of the art (lines 68-70 pag2).

Concerning data fusion applied for the analysis of mushrooms, I would suggest you provide a wider description of the state of the art, for instance, I am aware of another study about it, which could be mentioned:

-Yao, S.; Li, T.; Liu, H.G.; Li, J. Q.; Wang Y. Z. Geographic Characterization of Leccinum rugosiceps by Ultraviolet and Infrared Spectral Fusion. Anal Lett 2017, 50, 2257–2269.

And maybe some others are available in the literature; please check and in case enlarge the references including other studies on the same topic.

Section 2.4

The reference reported ([29]) is for sure very good. Nevertheless, mentioning a method, it would be nice to give reference also to the first author(s) who discussed it. Consequently, I would suggest you add at least the following references to PLS-DA:

- Sjöström, M.; Wold, S.; Söderström, B. PLS discriminant plots. In Pattern recognition in practice; Gelsema, E.S., Kanal L.N. Eds.; Elsevier: Amsterdam, Netherlands, 1986; pp. 461-470.

-Barker, M.; Rayens, W. Partial least squares for discrimination. J Chemometr 2003, 17, 166-173.

-Ståle, L.; Wold, S. Partial least squares analysis with cross-validation for the two-class problem: A Monte Carlo study. J Chemometr 1987, 1, 185-196.

The reference for VIP is missing, I would add at least the following one, in order to address the reader to the theory:

- Wold, S.; Johansson, E.; Cocchi. M. PLS: partial least squares projections to latent structures. In3D QSAR in Drug Design: Theory, Methods and Applications, H. Kubinyi Ed.; KLUWER ESCOM Science Publisher: Leiden, The Netherlands, 1993, pp. 523-550.

Response 5: Thanks for your suggestions. We have added the references into revised manuscript, and we have widened the state of the art in the introduction.

Point 6:

Line 117- page 3

The first time (in the manuscript) an acronym is mentioned in the text, please, report it together with the extended name. For instance, at line 117 of page 3 RSD is used; if it is not already present somewhere else into the manuscript, please provide the extended name.

Line 254 – page 9

Same comment as above. If I am not wrong, PCA is used for the first time at this point of the manuscript, but it the name is not reported in the extended form.

Response 6: We have supplemented the missing extended names in the revised manuscript.

Reviewer 2 Report

The study presents the geographical authentication of Macrohyporia cocos by a data fusion method combining ultra-fast liquid hromatography and FT-IR spectroscopy. The study is interesting, however there are some issues that, in my opinion, need to be addressed before publication.

The quality of Figure 1 is very poor. The peak at 2964 cm-1 is at the noise level. I think that the analysis based on this "peak" is ambiguous.

The quality of Figure 2. is also very poor, the structures of the compunds must be improved.

Were the inner parts and epidermis of each sample collected from the same M. cocos?

It would be nice to see some pictures (photograps) of the studied samples.

Table S2. Regression equations (a and values) are presented with too many significant digits.

It is not stated clearly how the content of each triterpene acid was determined. Spectrophotometrically?

Author Response

Response to reviewer2 comments

Point 1: The quality of Figure 1 is very poor. The peak at 2964 cm-1 is at the noise level. I think that the analysis based on this "peak" is ambiguous.

Response 1: Thanks for your suggestion. We have improved the quality of Figure 1. The peak located at 2964 cm-1 is unobvious in averaged derivative spectra, but it really belongs to M. cocos samples. As shown in the following figure that raw FTIR spectra of three Poriae Cutis samples, it showed clearly. Absorption at 2964 cm-1 correspond to C-H antisymmetric stretching vibration of methyl group. We have corrected the ambiguous statement in the revised manuscript.

Point 2: The quality of Figure 2. is also very poor, the structures of the compounds must be improved.

Response 2: We have made correction according to the reviewer’s comments.

Point 3: Were the inner parts and epidermis of each sample collected from the same M. cocos?

Response 3: The inner parts and epidermis of each sample were collected from the same M. cocos. In the process of data fusion, the LC and FTIR data of the same Poria sample were combined, and the FTIR data of inner parts and epidermis from the same sclerotium were combined. This content has been added in the Methods of revised manuscript.

Point 4: It would be nice to see some pictures (photographs) of the studied samples.

Response 4: Dried M. cocos sclerotium was showed below. We have added it in the revised manuscript.

Point 5: Table S2. Regression equations (a and values) are presented with too many significant digits.

Response 5: The number of effective digits has been reduced.

Point 6: It is not stated clearly how the content of each triterpene acid was determined. Spectrophotometrically?

Response 6: Contents of triterpene acids were determined by liquid chromatography. The description of determination has been added in Section 3.5 of revised manuscript.

“The developed UFLC method was validated in terms of precision, stability, repeatability, accuracy and linearity under the above chromatographic condition.

A mixed standard solution was determined six times successively within a day and on three consecutive days for evaluating intra- and inter-day precision. For the stability test, the extract of a sample was analyzed at 0, 6, 12, 17, 21 and 24 h, respectively. Six sample solutions prepared individually from the same sample were analyzed in parallel for evaluating the repeatability. The recovery test was performed to evaluate the accuracy by adding reference compounds of three different amounts (low, middle, and high) to the sample with known concentration accurately. The following equation was used to calculate recovery rate: Recovery rate (%) = [(measured amount - original amount) / spiked amount] × 100%.

The standard solutions of five compounds for constructing calibration curves were prepared by diluting the stock solutions with methanol individually. The ranges of concentration in the linearity study were 5.00–999 μg·mL-1 (dehydrotumulosic acid), 0.22–6730 μg·mL-1 (poricoic acid A), 2.4–480 μg·mL-1 (dehydropachymic acid), 10.3–1240 μg·mL-1 (pachymic acid) and 0.49–2450 μg·mL-1 (dehydrotrametenolic acid). Due to the obvious difference in contents of poricoic acid A of Poria and Poriae Cutis samples, two concentration ranges of 0.22–1121.95 μg·mL-1 (Poria) and 0.22–6730 μg·mL-1 (Poriae Cutis) were prepared. More than 7 levels (in arithmetic progression) of every concentration range were guaranteed. The limit of detection (LOD) and limit of quantification (LOQ) were determined by diluting continuously standard solution until the signal-to-noise ratios (S/N) reached about 3 and 10, respectively.”

Reviewer 3 Report

see attached.

Author Response

Point 1: Based on the introduction part and literature (which is well-written), I am not convinced about the novelty of the study. I recommend to the authors to provide some kind of comparison with the previous works in a table format or indicate which points of their research is new compared to the others.

Response 1: We have added it in revised manuscript.

“Infrared spectroscopy can provide the molecular functional group structure of metabolites. Liquid chromatography can characterize the exist of compounds and determinate the special compounds. They present different and complementary information, which were used to data fusion in this study. To the best of our knowledge, infrared spectroscopy was widely used for geographical classification because of the features of simplicity and rapidity [17,18]. Liquid chromatography was almost used for determining the contents of compounds [19,20]. Multiple chromatographic data fusion has been merely reported in the authentication of the geographical origin of palm oil [21], predicting antioxidant activity of Turnera diffusa [22], authentication of Valeriana species [23] as well as the comparison of Salvia miltiorrhiza and its variety [24]. Actually, wealthy information was contained in chromatographic data. And due to extensive automation, it could obtain a stable and reliable result.”

[17] Bureau, S.; Cozzolino, D.; Clark, C. J. Contributions of Fourier-transform mid infrared (FT-MIR) spectroscopy to the study of fruit and vegetables: A review. Postharvest Biol Tec 2019, 148, 1-14.

[18] Li, Y.; Zhang, J.; Wang, Y. FT-MIR and NIR spectral data fusion: a synergetic strategy for the geographical traceability of Panax notoginseng. Anal Bioanal Chem 2018, 410, 91-103.

[19] Wu, Z.; Zhao, Y.; Zhang, J.; Wang, Y. Quality assessment of Gentiana rigescens from different geographical origins using FT-IR spectroscopy combined with HPLC. Molecules 2017, 22, 1238.

[20] Wang, Y.; Shen, T.; Zhang, J.; Huang, H.; Wang, Y. Geographical authentication of Gentiana rigescens by high-performance liquid chromatography and infrared spectroscopy. Anal Lett 2018, 51, 2173-2191.

[21] Obisesan, K. A.; Jiménez-Carvelo, A. M.; Cuadros-Rodriguez, L.; Ruisánchez, I.; Callao, M. P. HPLC-UV and HPLC-CAD chromatographic data fusion for the authentication of the geographical origin of palm oil. Talanta 2017, 170, 413-418.

[22] Lucio-Gutiérrez, J. R.; Garza-Juárez, A.; Coello, J.; Maspoch, S.; Salazar-Cavazos, M. L.; Salazar-Aranda, R.; Waksman De Torres, N. Multi-wavelength high-performance liquid chromatographic fingerprints and chemometrics to predict the antioxidant activity of Turnera diffusa as part of its quality control. J Chromatogr a 2012, 1235, 68-76.

[23] Lucio-Gutiérrez, J. R.; Coello, J.; Maspoch, S. Enhanced chromatographic fingerprinting of herb materials by multi-wavelength selection and chemometrics. Anal Chim Acta 2012, 710, 40-49.

[24] Zhang, L.; Liu, Y.; Liu, Z.; Wang, C.; Song, Z.; Liu, Y.; Dong, Y.; Ning, Z.; Lu, A. Comparison of the roots of Salvia miltiorrhiza Bunge (Danshen) and its variety S. miltiorrhiza Bge f. Alba (Baihua Danshen) based on multi-wavelength HPLC-fingerprinting and contents of nine active components. Anal Methods-Uk 2016, 8, 3171-3182.

Point 2: In the materials and methods section I missed some important details. The number of samples at the first glance seems good based on Table 2, however later the authors confirmed the separation of the 78 samples into calibration and test set. I suggest to add some information about the distribution of the validation set. How many samples were in each group after the selection? The 26 samples all in all is a very small amount for the validation protocol, because it means that in equal distribution three samples / groups can be the average. It can cause some major problem with the models; they can be easily overfitted with the variables.

Response 2: We are sorry we have analyzed all the collected samples.

Point 3: The authors did not clarify which type of Q^2 was used? The definition of Q^2 is not unambiguous, see the paper [Viviana Consonni, Davide Ballabio and Roberto Todeschini, Evaluation of model predictive ability by external validation techniques, Journal of Chemometrics 2010; 24: 194-201.]

Response 3: Q2 was calculated based on leave-one-out cross-validation using objects of calibration set. We have written as “Q2(cum) represents the cross-validated cumulative R2, suggesting how well the model predicts new data” in revised manuscript.

Point 4: It is also interesting that the authors used VIP variable selection to prove some points based on the chromatograms, but they did not use it for variable selection at all? Did they try it as least? It would have been worth to try (even instead of the mid-level analysis), or any other variable selection technique especially if the number of samples is low.

Response 4: Thanks for your advice. We had tried to use VIP for variable selection, however, it did not show better performance than that of original variables. So, we did not use it for further analysis and did not write it into the manuscript. I’m sorry we can’t provide the parameters about model of variable selection.

Point 5: The results and discussion part are very superfluous, especially in section 2.5 and 2.6. There are too many details about the different models in each section, which has no additional information for the reader. They are just confusing. A figure or workflow would be helpful to summarize the details about the models somehow, or make some separation (lines, different font types, bold etc.) in Table 1.

Response 5: Thanks for your suggestion. We have added the workflow of geographical authentication using data fusion (Figure 5), which is helpful to understand how data was combined. We also add lines in Table 1. The figure and the table were shown below:

Figure 5.

Table 1.

Fusion Approach

Data Matrix

Calibration Set

Validation Set

R2(cum)

Q2(cum)

Accuracy

Accuracy

single technique

Poria

FTIR

0.8883

0.7268

100%

92.31%

LC242

0.6634

0.5277

96.15%

100%

LC210

0.5174

0.4012

90.38%

76.92%

Poria Cutis

FTIR

0.9292

0.6981

100%

96.15%

LC242

0.2874

0.2204

65.38%

34.62%

low-level data fusion

Poria

FTIR-LC242

0.9599

0.7917

100%

100%

FTIR-LC210

0.9468

0.7663

100%

100%

LC242-210

0.8097

0.6547

98.08%

92.31%

FTIR-LC242-210

0.8823

0.7566

100%

100%

Poria Cutis

FTIR-LC242

0.9016

0.7032

100%

100%

FTIR-LC242-210

0.905

0.698

100%

100%

combination data of two medicinal parts

FTIR

0.9548

0.8064

100%

100%

LC242

0.8147

0.6495

100%

100%

LC210

0.6489

0.4806

94.23%

88.46%

mid-level data fusion

Poria

FTIR-LC242

0.8266

0.5745

100%

100%

FTIR-LC210

0.7453

0.5053

96.15%

96.15%

FTIR-LC242-210

0.8286

0.5882

100%

100%

Poria Cutis

FTIR-LC242

0.7386

0.5493

100%

92.31%

FTIR-LC210

0.7518

0.4991

100%

96.15%

LC242-210

0.4617

0.228

76.92%

73.08%

FTIR-LC242-210

0.7607

0.5558

100%

96.15%

combination data of two medicinal parts

FTIR

0.7564

0.5982

98.08%

88.46%

LC242

0.7761

0.4973

98.08%

100%

LC210

0.676

0.3756

96.15%

88.46%

Point 6: Table 1 clearly shows that something is strange and not normal with the models. The accuracies seem pretty good in almost every case, however the differences between R^2 and Q^2 values are usually very big. It is not a good sign, the authors can easily verify these results based on the work [Toth, G. et al., J. Comput. Aided Mol. Des. 27 (2013) 837-844.]. The ratio of the two parameters can be calculated by (1-R^2)/(1-Q^2).

Thus the models seem to be overfitted at least. I recommend adding more validation procedures to the study, select the validation set in a proper way (Kennard-Stone is over-optimistic). At least the best models should be modified for example with less latent variables (or less variables), where the R^2 and Q^2 values are closer to each other. Sometimes less is more, even if the accuracy is sacrificed. On the other hand, there are more cross-validation options that the authors can apply to verify their models more thoroughly. Please select the best models and put them into a separate Table, or indicate it in Table 1, otherwise the readers get lost in the bunch of data.

Response 6: Thanks for your advice. We have validated PLS-DA models using permutation test.

“To validate the performance of the PLS‐DA model, a 30‐iteration permutation test was performed. As shown in Figure S2 that one of permutations plots for Poria LC242-210 model, all permutated Q2 and R2 values (bottom left) were lower than the corresponding original values (top right). It indicated that the PLS-DA model was considered as an appropriate model without random and overfitting. The results showed that all the PLS-DA models were not overfitting.”

Figure S2.

Point 7: Figure 3 is not explained well in the caption. Please indicate which chromatogram is which type of sample. And please explain the difference in the case of Figure 3B.

Response 7: We have written as “Figure 3. Chromatograms of Poria (A, B) and Poriae Cutis (C, D) recorded at 242 (A, C) and 210 nm (B, D) after the transformation of correlation optimized warping”. Figure 3B is chromatograms of Poria samples at 210 nm, whose baseline is not flat compared with those at 242 nm. It may be that the change of mobile phase was easily shown at 210 nm. Because the chromatogram of blank control group (following figure) had similar trend.

Figure.

Point 8: Section 2.2: RSD should be written in full length at the first usage, even if it is obvious.

Response 8: We have supplemented it in revised manuscript.

Point 9: Section 2.2: Please shorten the results of ANOVA method. It is superfluous. The main information should be emphasized in a shorter way.

Response 9: We have made correction according to the reviewer’s comments. The results of ANOVA method are shown below:

“Poricoic acid A contents of Mengmeng were significantly different from those of Beicheng, Tuodian and Zhanhe in inner parts, and Yongping in cutis samples. Contents of dehydropachymic acid and pachymic acid in inner parts from Caodian were higher than those of other geographical origins except for Baliu. Inner parts from Baliu showed higher contents of dehydropachymic acid than those from Beicheng, Dawen and Mengmeng, and higher contents of pachymic acid than those from Tuodian, Yongping, Beicheng and Mengmeng. Inner parts from Dawen contained fairly low contents of dehydrotrametenolic acid compared with those from others with the exception of Baliu. Compared with epidermis samples from Dawen, Beicheng and Yongping showed higher contents of dehydrotumulosic acid, and Caodian and Baliu presented higher amount of pachymic acid.”

Point 10: Figure S2 should be transformed to the materials and methods section, it helps a lot to understand what it means.

Response 2: We have made correction according to the reviewer’s comments.

Reviewer 4 Report

After reading the manuscript I have the following remarks and suggestions:

Elements of scientific novelty should be presented in a more detailed and convincing manner (in the last paragraph of the Introduction).

Innovative potential of the results obtained should be explained in detail (CONCLUSIONS).

Application of proper quality assurance/quality control (QA/QC) procedures is vital for the measurement results to be treated as a source of reliable analytical information. Consequently, I suggest that a separate section devoted to QA/QC be added to the manuscript. Special attention should be paid to:

- description of the validation procedure for the applied/proposed analytical protocol,

- information on metrological characteristics of the analytical procedure, especially Method Quantitation Limit (MQL) values for the entire procedure (from handling of representative samples to statistical and chemometric evaluation of the data sets obtained), and not only for the analytical techniques used during the analysis of the extracts.

Advantages and drawbacks of the proposed aproach should be presented.

Author Response

Point 1: Elements of scientific novelty should be presented in a more detailed and convincing manner (in the last paragraph of the Introduction).

Response 1: According to the reviewer’s comments, we have made correction in the last paragraph of the Introduction.

In this study, two data fusion strategies including low- and mid-level fusion as well as two data combinations including the fusion of complementary information of single part and the fusion of information of two medicinal parts from one sclerotium were performed for geographical authentication of M. cocos. Liquid chromatograms at two wavelengths (242 and 210 nm) and FTIR spectra of two medicinal parts (Poria and Poriae Cutis) of M. cocos were analyzed. Contents of five triterpene acids were measured. Chromatographic data fusion, spectral data fusion as well as chromatography and spectroscopy data fusion were implemented combined with partial least squares discriminant analysis (PLS-DA).”

Point 2: Innovative potential of the results obtained should be explained in detail (CONCLUSIONS).

Response 2: We have made correction in the first paragraph of the Conclusions.

“In order to establish an effective method for geographical authentication of M. cocos, two data fusion strategies including low- and mid-level fusion as well as two data combinations including the fusion of complementary information of single part and the fusion of information of two parts from one sclerotium were compared. FTIR, LC242 and LC210 were used to characterize the epidermis and inner part of M. cocos sclerotium from different places individually and jointly.”

Point 3: Application of proper quality assurance/quality control (QA/QC) procedures is vital for the measurement results to be treated as a source of reliable analytical information. Consequently, I suggest that a separate section devoted to QA/QC be added to the manuscript. Special attention should be paid to:

- description of the validation procedure for the applied/proposed analytical protocol,

- information on metrological characteristics of the analytical procedure, especially Method Quantitation Limit (MQL) values for the entire procedure (from handling of representative samples to statistical and chemometric evaluation of the data sets obtained), and not only for the analytical techniques used during the analysis of the extracts.

Response 3: Thanks for your suggestion, this part has been added in Section 3.5 of revised manuscript. The repeated sentences in the other section have been deleted.

3.5. Method validation

The developed UFLC method was validated in terms of precision, stability, repeatability, accuracy and linearity under the above chromatographic condition.

A mixed standard solution was determined six times successively within a day and on three consecutive days for evaluating intra- and inter-day precision. For the stability test, the extract of a sample was analyzed at 0, 6, 12, 17, 21 and 24 h, respectively. Six sample solutions prepared individually from the same sample were analyzed in parallel for evaluating the repeatability. The recovery test was performed to evaluate the accuracy by adding reference compounds of three different amounts (low, middle, and high) to the sample with known concentration accurately. The following equation was used to calculate recovery rate: Recovery rate (%) = [(measured amount - original amount) / spiked amount] × 100%.

The standard solutions of five compounds for constructing calibration curves were prepared by diluting the stock solutions with methanol individually. The ranges of concentration in the linearity study were 5.00–999 μg·mL-1 (dehydrotumulosic acid), 0.22–6730 μg·mL-1 (poricoic acid A), 2.4–480 μg·mL-1 (dehydropachymic acid), 10.3–1240 μg·mL-1 (pachymic acid) and 0.49–2450 μg·mL-1 (dehydrotrametenolic acid). Due to the obvious difference in contents of poricoic acid A of Poria and Poriae Cutis samples, two concentration ranges of 0.22–1121.95 μg·mL-1 (Poria) and 0.22–6730 μg·mL-1 (Poriae Cutis) were prepared. More than 7 levels (in arithmetic progression) of every concentration range were guaranteed. The limit of detection (LOD) and limit of quantification (LOQ) were determined by diluting continuously standard solution until the signal-to-noise ratios (S/N) reached about 3 and 10, respectively.”

Point 4: Advantages and drawbacks of the proposed approach should be presented.

Response 4: In this study, our proposed approach mainly takes use of data fusion, so the drawback is that we have to collect multiple kinds of data if we want to use it, which means more time and effort will be taken. The advantage is that it has more opportunity to get an accurate result compared with independent technique. This part was presented as “although time and effort would be taken to collect multiple complementary data, the data fusion was suggested as an alternative strategy to show accurate characterization” in the Introduction.

Reviewer 5 Report

Wang et al. have submitted a manuscript detailing geographical authentication of Macrohyporia cocos using an approach of data fusion combining LC and FTIR data resulting in PLS-DA strongly predictive classification models. The study is interesting and of interest to the readers of Molecules. However, minor revisions, as detailed below are needed prior to its potential acceptance for publication.

1. Does LC242 and LC210 refer to 242 and 210 nm? Clarify this in the abstract.

2. FTIR and other spectroscopic techniques are widely used for geographical classification. The  wealth information contained in chromatographic data is vast and LC is nowadays widely used in bionalysis [1]. However, in the Introduction is not entirely clear what is the rationale of using chromatographic data for geographical classification (authentication).

3. A more thorough literature overview must be given in the Introduction covering other such studies to further justify this work. Moreover, a clear distinction must be given between this work and the authors' previous studies (e.g., Wang et al. Anal. Lett. 2018 [2]; Wu et al. Molecules 2017 [3]). The study seems to detail quite an incremental advance w.r.t. previous studies. Please strengthen and justify the latest work from your research group.

4. It is not entirely sure what was used as predictors in the PLS-DA models. For FTIR it is the spectra; but for the LC data, is it the whole chromatograms, or only retention times of the peak apexes?

5. Since the goal of the study is to utilize both IR & chromatographic data, why not perform HPLC with FTIR instead of UV detection (such as e.g., Robb et al. J. Liq. Chrom. Rel. Technol 2002 [4])?

6. How was the gradient elution profile optimized? Was it by trial-and-error?

7. Figure 1 is illegible. Increase its quality. Especially the text.

8. For Figure 2, same comment as for Figure 1. Improve the quality.

9. For Figure 3, same comment as for Figures 1 & 2.

10. Lines 338-339: "data dinning"?

11. Lines 383-385: not entirely correct. Cumulative R2 represents the percentage of explained variance for a defined number of LVs, while Q2 represents the cross-validated R2.

[1] Žuvela, P.; Skoczylas, M.; Liu, J. J.; Baczek, T.; Kaliszan, R.; Wong, M. W.; Buszewski, B. Column Characterization and Selection Systems in Reversed-Phase High-Performance Liquid Chromatography. Chem. Rev. 2019

[2] Wang, Y.; Shen. T.; Zhang, J.; Huang, H. Y.; Wang, Y. Geographical Authentication of Gentiana Rigescens by High-Performance Liquid Chromatography and Infrared Spectroscopy. Anal. Lett. 2018, 51, 1-19.

[3] Wu, Z.; Zhao, Y.; Zhang, J.; Wang, Y. Quality Assessment of Gentiana rigescens from Different Geographical Origins Using FT-IR Spectroscopy Combined with HPLC. Molecules 2017, 22, 1238.

[4] Robb; S. C.; Geldart, S. E.; Seelenbinder, J. A.; Brown, P. R. Analysis of green tea constituents by HPLC-FTIR. J. Liq. Chrom. Rel. Technol. 2002, 25, 787-801.

Author Response

Point 1: Does LC242 and LC210 refer to 242 and 210 nm? Clarify this in the abstract.

Response 1: Thanks for your advice. This has been clarified in the abstract.

Point 2: FTIR and other spectroscopic techniques are widely used for geographical classification. The wealth information contained in chromatographic data is vast and LC is nowadays widely used in bionalysis [1]. However, in the Introduction is not entirely clear what is the rationale of using chromatographic data for geographical classification (authentication).

[1] Žuvela, P.; Skoczylas, M.; Liu, J. J.; Baczek, T.; Kaliszan, R.; Wong, M. W.; Buszewski, B. Column Characterization and Selection Systems in Reversed-Phase High-Performance Liquid Chromatography. Chem. Rev. 2019

Response 2: Thanks for your advice. We have described these in the Introduction. The reason why using chromatographic data for geographical classification is that liquid chromatography can characterize the exist of compounds and determinate the special compounds. Due to extensive automation, it can obtain a stable and reliable result. When all of samples are analyzed using the same LC method, the chemical characteristics of samples from different geographic origins can be showed.

“Infrared spectroscopy can provide the molecular functional group structure of metabolites. Liquid chromatography can characterize the exist of compounds and determinate the special compounds. They present different and complementary information, which were used to data fusion in this study. To the best of our knowledge, infrared spectroscopy was widely used for geographical classification because of the features of simplicity and rapidity [17,18]. Liquid chromatography was almost used for determining the contents of compounds [19,20]. Multiple chromatographic data fusion has been merely reported in the authentication of the geographical origin of palm oil [21], predicting antioxidant activity of Turnera diffusa [22], authentication of Valeriana species [23] as well as the comparison of Salvia miltiorrhiza and its variety [24]. Actually, wealthy information was contained in chromatographic data. And due to extensive automation, it could obtain a stable and reliable result.”

[17] Bureau, S.; Cozzolino, D.; Clark, C. J. Contributions of Fourier-transform mid infrared (FT-MIR) spectroscopy to the study of fruit and vegetables: A review. Postharvest Biol Tec 2019, 148, 1-14.

[18] Li, Y.; Zhang, J.; Wang, Y. FT-MIR and NIR spectral data fusion: a synergetic strategy for the geographical traceability of Panax notoginseng. Anal Bioanal Chem 2018, 410, 91-103.

[19] Wu, Z.; Zhao, Y.; Zhang, J.; Wang, Y. Quality assessment of Gentiana rigescens from different geographical origins using FT-IR spectroscopy combined with HPLC. Molecules 2017, 22, 1238.

[20] Wang, Y.; Shen, T.; Zhang, J.; Huang, H.; Wang, Y. Geographical authentication of Gentiana rigescens by high-performance liquid chromatography and infrared spectroscopy. Anal Lett 2018, 51, 2173-2191.

[21] Obisesan, K. A.; Jiménez-Carvelo, A. M.; Cuadros-Rodriguez, L.; Ruisánchez, I.; Callao, M. P. HPLC-UV and HPLC-CAD chromatographic data fusion for the authentication of the geographical origin of palm oil. Talanta 2017, 170, 413-418.

[22] Lucio-Gutiérrez, J. R.; Garza-Juárez, A.; Coello, J.; Maspoch, S.; Salazar-Cavazos, M. L.; Salazar-Aranda, R.; Waksman De Torres, N. Multi-wavelength high-performance liquid chromatographic fingerprints and chemometrics to predict the antioxidant activity of Turnera diffusa as part of its quality control. J Chromatogr a 2012, 1235, 68-76.

[23] Lucio-Gutiérrez, J. R.; Coello, J.; Maspoch, S. Enhanced chromatographic fingerprinting of herb materials by multi-wavelength selection and chemometrics. Anal Chim Acta 2012, 710, 40-49.

[24] Zhang, L.; Liu, Y.; Liu, Z.; Wang, C.; Song, Z.; Liu, Y.; Dong, Y.; Ning, Z.; Lu, A. Comparison of the roots of Salvia miltiorrhiza Bunge (Danshen) and its variety S. miltiorrhiza Bge f. Alba (Baihua Danshen) based on multi-wavelength HPLC-fingerprinting and contents of nine active components. Anal Methods-Uk 2016, 8, 3171-3182.

Point 3: A more thorough literature overview must be given in the Introduction covering other such studies to further justify this work. Moreover, a clear distinction must be given between this work and the authors' previous studies (e.g., Wang et al. Anal. Lett. 2018 [2]; Wu et al. Molecules 2017 [3]). The study seems to detail quite an incremental advance w.r.t. previous studies. Please strengthen and justify the latest work from your research group.

[2] Wang, Y.; Shen. T.; Zhang, J.; Huang, H. Y.; Wang, Y. Geographical Authentication of Gentiana Rigescens by High-Performance Liquid Chromatography and Infrared Spectroscopy. Anal. Lett. 2018, 51, 1-19.

[3] Wu, Z.; Zhao, Y.; Zhang, J.; Wang, Y. Quality Assessment of Gentiana rigescens from Different Geographical Origins Using FT-IR Spectroscopy Combined with HPLC. Molecules 2017, 22, 1238.

Response 3: Thanks for your advice. We have corrected these in revised manuscript. Our research group had used liquid chromatography and infrared spectroscopy. However, we just utilized LC for determination of active components contents and did not take full use of LC data, namely, the whole chromatograms. At the same time, it is the first time to combine LC with FTIR by data fusion strategies for geographical classification of M. cocos.

Point 4: It is not entirely sure what was used as predictors in the PLS-DA models. For FTIR it is the spectra; but for the LC data, is it the whole chromatograms, or only retention times of the peak apexes?

Response 4: The whole chromatograms was used. We have described in the revised manuscript.

Point 5: Since the goal of the study is to utilize both IR & chromatographic data, why not perform HPLC with FTIR instead of UV detection (such as e.g., Robb et al. J. Liq. Chrom. Rel. Technol 2002 [4])?

[4] Robb; S. C.; Geldart, S. E.; Seelenbinder, J. A.; Brown, P. R. Analysis of green tea constituents by HPLC-FTIR. J. Liq. Chrom. Rel. Technol. 2002, 25, 787-801.

Response 5: Thanks for your advice. FTIR detection is suited to the detection of triterpene compounds due to their high infrared activity. HPLC-FTIR can get the spectra of components which eluted at different retention times, which can be used to qualitatively identify triterpenes in M. cocos extracts and get detailed information. However, we collected the infrared spectrum of M. cocos powder and paid attention to the comprehensive information in this study. If we have this equipment in the future, we can try which one works better for geographical authentication of M. cocos.

Point 6: How was the gradient elution profile optimized? Was it by trial-and-error?

Response 6: The gradient elution profile was optimized by trial-and-error on the basis of literatures and our experience.

Point 7: Figure 1 is illegible. Increase its quality. Especially the text.

For Figure 2, same comment as for Figure 1. Improve the quality.

For Figure 3, same comment as for Figures 1 & 2.

Lines 338-339: "data dinning"?

Response 7: We are very sorry for our negligence. We have improved the quality of figures, and the word group has been rewritten as “data binning” in the revised manuscript.

Point 8: Lines 383-385: not entirely correct. Cumulative R2 represents the percentage of explained variance for a defined number of LVs, while Q2 represents the cross-validated R2.

Response 8: Considering the reviewer’s comment, we have rewritten as “R2(cum) represents the percentage of explained variance for a defined number of latent variables, indicating how well the model fits the data. Q2(cum) represents the cross-validated cumulative R2, suggesting how well the model predicts new data”.

Round 2

Reviewer 2 Report

The Authors have answered most of my questions.

Reviewer 3 Report

I accept the manuscript in its present form.

Reviewer 4 Report

After the lecture of the revised version of the manuscript and reflexion. I can state that I have no additional remarks ans/or comments